# ARC - Actor Residual Critic
# for Adversarial Imitation Learning

**Ankur Deka**[†*], **Changliu Liu**[†], **Katia Sycara**[†]
[†]Robotics Institute, Carnegie Mellon University    [*]Intel Labs
adeka@alumni.cmu.edu, {cliu6,katia}@cs.cmu.edu

**Abstract:** Adversarial Imitation Learning (AIL) is a class of popular state-of-the-art Imitation Learning algorithms commonly used in robotics. In AIL, an artificial adversary's misclassification is used as a reward signal that is optimized by any standard Reinforcement Learning (RL) algorithm. Unlike most RL settings, the reward in AIL is *differentiable* but current model-free RL algorithms do not make use of this property to train a policy. The reward is AIL is also *shaped* since it comes from an adversary. We leverage the differentiability property of the shaped AIL reward function and formulate a class of Actor Residual Critic (ARC) RL algorithms. ARC algorithms draw a parallel to the standard Actor-Critic (AC) algorithms in RL literature and uses a residual critic, $C$ function (instead of the standard $Q$ function) to approximate only the discounted future return (excluding the immediate reward). ARC algorithms have similar convergence properties as the standard AC algorithms with the additional advantage that the gradient through the immediate reward is exact. For the discrete (tabular) case with finite states, actions, and known dynamics, we prove that policy iteration with $C$ function converges to an optimal policy. In the continuous case with function approximation and unknown dynamics, we experimentally show that ARC aided AIL outperforms standard AIL in simulated continuous-control and real robotic manipulation tasks. ARC algorithms are simple to implement and can be incorporated into any existing AIL implementation with an AC algorithm. Video and link to code are available at: `sites.google.com/view/actor-residual-critic`.

**Keywords:** Adversarial Imitation Learning (AIL), Actor-Critic (AC), Actor Residual Critic (ARC)

## 1   Introduction

Although Reinforcement Learning (RL) allows us to train agents to perform complex tasks without manually designing controllers [1, 2, 3], it is often tedious to hand-craft a dense reward function that captures the task objective in robotic tasks [4, 5, 6]. Imitation Learning (IL) or Learning from Demonstration (LfD) is a popular choice in such situations [4, 5, 6, 7]. Common approaches to IL are Behavior Cloning (BC) [8] and Inverse Reinforcement Learning (IRL) [9].

Within IRL, recent Adversarial Imitation Learning (AIL) algorithms have shown state-of-the-art performance, especially in continuous control tasks which make them relevant to real-world robotics problems. AIL methods cast the IL problem as an adversarial game between a policy and a learned adversary (discriminator). The adversary aims to classify between agent and expert trajectories and the policy is trained using the adversary's mis-classification as the reward function. This encourages the policy to imitate the expert. Popular AIL algorithms include Generative Adversarial Imitation Learning (GAIL) [10], Adversarial Inverse Reinforcement Learning (AIRL) [11] and $f$-MAX [12].

The agent in AIL is trained with any standard RL algorithm. There are two popular categories of RL algorithms: (i) on-policy algorithms such as TRPO [13], PPO [2], GAE [14] based on the policy gradient theorem [15, 16]; and (ii) off-policy Actor-Critic (AC) algorithms such as DDPG [17], TD3 [18], SAC [3] that compute the policy gradient through a critic ($Q$ function). These standard RL algorithms were designed for arbitrary scalar reward functions; and they compute an *approximate* gradient for updating the policy. Practical on-policy algorithms based on the policy gradient theorem use several approximations to the true gradient [13, 2, 14] and off-policy AC

6th Conference on Robot Learning (CoRL 2022), Auckland, New Zealand.

algorithms first approximate policy return with a critic ($Q$ function) and subsequently compute the gradient through this critic [17, 18, 3]. Even if the $Q$ function is approximated very accurately, the error in its gradient can be arbitrarily large, Appendix A.1.

Our insight is that the reward function in AIL has 2 special properties: (i) it is *differentiable* which means we can compute the exact gradient through the reward function instead of approximating it and (ii) it is dense/shaped as it comes from an adversary. As we will see in section 3, naively computing the gradient through reward function would lead to a short-sighted sub-optimal policy. To address this issue, we formulate a class of Actor Residual Critic (ARC) RL algorithms that use a residual critic, $C$ function (instead of the standard $Q$ function) to approximate only the discounted future return (excluding immediate reward).

The contribution of this paper is the introduction of ARC, which can be easily incorporated to replace the AC algorithm in any existing AIL algorithm for continuous-control and helps boost the asymptotic performance by computing the exact gradient through the shaped reward function.

## 2   Related Work

| Algorithm | Minimized $f$-Divergence | | $r(s,a)$ |
|---|---|---|---|
| | **Name** | **Expression** | |
| GAIL [10] | Jensen-Shannon | $\frac{1}{2}\left\{\mathbb{E}_{\rho^{\exp}}\log\frac{2\rho^{\exp}}{\rho^{\exp}+\rho^{\pi}}+\mathbb{E}_{\rho^{\pi}}\log\frac{2\rho^{\pi}}{\rho^{\exp}+\rho^{\pi}}\right\}$ | $\log D(s,a)$ |
| AIRL [11], $f$-MAX-RKL [12] | Reverse KL | $\mathbb{E}_{\rho^{\pi}}\log\frac{\rho^{\pi}}{\rho^{\exp}}$ | $\log\frac{D(s,a)}{1-D(s,a)}$ |

Table 1: Popular AIL algorithms, $f$-divergence metrics they minimize and their reward functions.

The simplest approach to imitation learning is Behavior Cloning [8] where an agent policy directly regresses on expert actions (but not states) using supervised learning. This leads to distribution shift and poor performance at test time [19, 10]. Methods such as DAgger [19] and Dart [20] eliminate this issue but assume an interactive access to an expert policy, which is often impractical.

Inverse Reinforcement Learning (IRL) approaches recover a reward function which can be used to train an agent using RL [9, 21] and have been more successful than BC. Within IRL, recent Adversarial Imitation Learning (AIL) methods inspired by Generative Adversarial Networks (GANs) [22] have been extremely successful. GAIL [10] showed state-of-the-art results in imitation learning tasks following which several extensions have been proposed [23, 24]. AIRL [11] imitates an expert as well as recovers a robust reward function. [25] and [12] presented a unifying view on AIL methods by showing that they minimize different divergence metrics between expert and agent state-action distributions but are otherwise similar. [12] also presented a generalized AIL method $f$-MAX which can minimize any specified $f$-divergence metric [26] between expert and agent state-action distributions thereby imitating the expert. Choosing different divergence metrics leads to different AIL algorithms, e.g. choosing Jensen-Shannon divergence leads to GAIL [10]. [27] proposed a method that automatically learns a $f$-divergence metric to minimize. Our proposed Actor Residual Critic (ARC) can be augmented with any of these AIL algorithms to leverage the reward gradient.

Some recent methods have leveraged the differentiable property of reward in certain scenarios but they have used this property in very different settings. [28] used the gradient of the reward to improve the reward function but not to optimize the policy. We on the other hand explicitly use the gradient of the reward to optimize the policy. [29] used the gradient through the reward to optimize the policy but operated in the model-based setting. If we have access to a differentiable dynamics model, we can directly obtain the gradient of the expected return (policy objective) w.r.t. the policy parameters, Appendix E.5. Since we can directly obtain the objective's gradient, we do not necessarily need to use either a critic ($Q$) as in standard Actor Critic (AC) algorithms or a residual critic ($C$) as in our proposed Actor Residual Critic (ARC) algorithms. Differentiable cost (negative reward) has also been leveraged in control literature for a long time to compute a policy, e.g. in LQR [30] and its extensions; but they assume access to a known dynamics model. We on the other hand present a model-free method with unknown dynamics that uses the gradient of the reward to optimize the policy with the help of a new class of RL algorithms called Actor Residual Critic (ARC).

## 3   Background

**Objective**   Our goal is to imitate an expert from one or more demonstrated trajectories (state-action sequences) in a continuous-control task (state and action spaces are continuous). Given any Adver-

sarial Imitation Learning (AIL) algorithm that uses an off-policy Actor-Critic algorithm RL algorithm, we wish to use our insight on the availability of a differentiable reward function to improve the imitation learning algorithm.

**Notation** The environment is modeled as a Markov Decision Process (MDP) represented as a tuple $(\mathcal{S}, \mathcal{A}, \mathcal{P}, r, \rho_0, \gamma)$ with state space $\mathcal{S}$, action space $\mathcal{A}$, transition dynamics $\mathcal{P} : \mathcal{S} \times \mathcal{A} \times \mathcal{S} \to [0, 1]$, reward function $r(s, a)$, initial state distribution $\rho_0(s)$, and discount factor $\gamma$. $\pi(.|s)$, $\pi^{\exp}(.|s)$ denote policies and $\rho^\pi, \rho^{\exp} : \mathcal{S} \times \mathcal{A} \to [0, 1]$ denote state-action occupancy distributions for agent and expert respectively. $\mathcal{T} = \{s_1, a_1, s_2, a_2, \ldots, s_T, a_T\}$ denotes a trajectory or episode and $(s, a, s', a')$ denotes a continuous segment in a trajectory. A discriminator or adversary $D(s, a)$ tries to determine whether the particular $(s, a)$ pair belongs to an expert trajectory or agent trajectory, i.e. $D(s, a) = P(\text{expert}|s, a)$. The optimal discriminator is $D(s, a) = \frac{\rho^{\exp}(s,a)}{\rho^{\exp}(s,a)+\rho^\pi(s,a)}$ [22].

**Adversarial Imitation Learning (AIL)** In AIL, the discriminator and agent are alternately trained. The discriminator is trained to maximize the likelihood of correctly classifying expert and agent data using supervised learning, (1) and the agent is trained to maximize the expected discounted return, (2).

$$\max_D \left\{ \mathbb{E}_{s,a\sim\rho^{\exp}}[\log D(s, a)] + \mathbb{E}_{s,a\sim\rho^\pi}[\log(1 - D(s, a))] \right\} \tag{1}$$

$$\max_\pi \left\{ \mathbb{E}_{s,a\sim\rho_0,\pi,\mathcal{P}} \sum_{t\geq0} \gamma^t r(s_t, a_t) \right\} \tag{2}$$

Here, reward $r_\psi(s, a) = h(D_\psi(s, a))$ is a function of the discriminator which varies between different AIL algorithms. Different AIL algorithms minimize different $f$-divergence metrics between expert and agent state-action distribution. Defining a $f$-divergence metric instantiates different reward functions [12]. Some popular divergence choices are Jensen-Shannon in GAIL [10] and Reverse Kullback-Leibler in $f$-MAX-RKL [12] and AIRL [11] as shown in Table 1.

Any RL algorithm could be used to optimize (2) and popular choices are off-policy Actor-Critic algorithms such as DDPG [17], TD3 [18], SAC [3] and on-policy algorithms such as TRPO [13], PPO [2], GAE [14] which are based on the policy gradient theorem [15, 16]. We focus on off-policy Actor-Critic algorithms as they are usually more sample efficient and stable than on-policy policy gradient algorithms [18, 3].

**Continuous-control using off-policy Actor-Critic** The objective in off-policy RL algorithms is to maximize expected $Q$ function of the policy, $Q^\pi$ averaged over the state distribution of a dataset $\mathcal{D}$ (typically past states stored in buffer) and the action distribution of the policy $\pi$ [31]:

$$\max_\pi \mathbb{E}_{s\sim\mathcal{D},a\sim\pi} Q^\pi(s, a) \tag{3}$$

$$\text{where, } Q^\pi(s, a) = \mathbb{E}_{s,a\sim\rho_0,\pi,\mathcal{P}} \left[ \sum_{k\geq0} \gamma^k r_{t+k} \middle| s_t = s, a_t = a \right] \tag{4}$$

The critic and the policy denoted by $Q$, $\pi$ respectively are approximated by function approximators such as neural networks with parameters $\phi$ and $\theta$ respectively. There is an additional target $Q_{\phi_{\text{targ}}}$ function parameterized by $\phi_{\text{targ}}$. There are two alternating optimization steps:

1. Policy evaluation: Fit critic ($Q_\phi$ function) by minimizing Bellman Backup error.

$$\min_\phi \mathbb{E}_{s,a,s'\sim\mathcal{D}} \left\{ Q_\phi(s, a) - y(s, a) \right\}^2 \tag{5}$$

$$\text{where, } y(s, a) = r(s, a) + \gamma Q_{\phi_{\text{targ}}}(s', a') \text{ and } a' \sim \pi_\theta(.|s') \tag{6}$$

$Q_\phi$ is updated with gradient descent without passing gradient through the target $y(s, a)$.

2. Policy improvement: Update policy with gradient ascent over RL objective.

$$\mathbb{E}_{s\sim\mathcal{D}} \left[ \nabla_\theta Q_\phi(s, a \sim \pi_\theta(.|s)) \right] \tag{7}$$

All off-policy Actor Critic algorithms follow the core idea above ((5) and (7)) along with additional details such as the use of a deterministic policy and target network in DDPG [17], double Q networks and delayed updates in TD3 [18], entropy regularization and reparameterization trick in SAC [3].

**Naive-Diff and why it won't work**  Realizing that the reward in AIL is differentiable and shaped, we can formulate a Naive-Diff RL algorithm that updates the policy by differentiating the RL objective (2) with respect to the policy parameters $\theta$.

$$\mathbb{E}_{\mathcal{T}\sim\mathcal{D}}\left[\nabla_\theta r(s_1,a_1) + \gamma\nabla_\theta r(s_2,a_2) + \gamma^2\nabla_\theta r(s_3,a_3) + \dots\right] \tag{8}$$

$\mathcal{T} = \{s_1,a_1,s_2,a_2\dots\}$ is a sampled trajectory in $\mathcal{D}$. Using standard autodiff packages such as Pytorch [32] or Tensorflow [33] to naively compute the gradients in (8) would produce incorrect gradients. Apart from the immediate reward $r(s_1,a_1)$, all the terms depend on the transition dynamics of the environment $\mathcal{P}(s_{t+1}|s_t,a_t)$, which is unknown and we cannot differentiate through it. So, autodiff will calculate the gradient of only immediate reward correctly and calculate the rest as 0's. This will produce a short-sighted sub-optimal policy that maximizes only the immediate reward.

## 4  Method

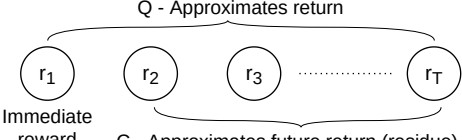

Figure 1: Visual illustration of approximating reward via Q function or C function.

The main lesson we learnt from Naive-Diff is that while we can obtain the gradient of immediate reward, we cannot directly obtain the gradient of future return due to unknown environment dynamics. This directly motivates our formulation of Actor Residual Critic (ARC). Standard Actor Critic algorithms use $Q$ function to approximate the return as described in Eq. 4. However, since we can directly obtain the gradient of the reward, we needn't approximate it with a $Q$ function. We, therefore, propose to use $C$ function to approximate only the future return, leaving out the immediate reward. This is the core idea behind Actor Residual Critic (ARC) and is highlighted in Fig. 1. The word "Residual" refers to the amount of return that remains after subtracting the immediate reward from the return. As we will see in Section 4.3, segregating the immediate reward from future return will allow ARC algorithms to leverage the exact gradient of the shaped reward. We now formally describe Residual Critic ($C$ function) and its relation to the standard critic ($Q$ function).

### 4.1  Definition of Residual Critic ($C$ function)

The Q function under a policy $\pi$, $Q^\pi(s,a)$, is defined as the expected discounted return from state $s$ taking action $a$, (9). The $C$ function under a policy $\pi$, $C^\pi(s,a)$, is defined as the expected discounted future return, excluding the immediate reward (10). Note that the summation in (10) starts from 1 instead of 0. $Q$ function can be expressed in terms of $C$ function as shown in (11).

$$Q^\pi(s,a) = \mathbb{E}_{s,a\sim\rho_0,\pi,\mathcal{P}}\left[\sum_{k\geq0}\gamma^k r_{t+k}\bigg|s_t=s,a_t=a\right] \tag{9}$$

$$C^\pi(s,a) = \mathbb{E}_{s,a\sim\rho_0,\pi,\mathcal{P}}\left[\sum_{k\geq1}\gamma^k r_{t+k}\bigg|s_t=s,a_t=a\right] \tag{10}$$

$$Q^\pi(s,a) = r(s,a) + C^\pi(s,a) \tag{11}$$

### 4.2  Policy Iteration using $C$ function

---

**Algorithm 1:** Policy Iteration with $C$ function

---

Initialize $C^0(s,a)\forall s,a$;
**while** $\pi$ *not converged* **do**
  // Policy evaluation
  **for** *n=1,2,… until $C_k$ converges* **do**
  | $C^{n+1}(s,a) \leftarrow \gamma\sum_{s'}P(s'|s,a)\sum_{a'}\pi(a'|s')\left(r(s',a')+C^n(s',a')\right)$   $\forall s,a$
  // Policy improvement
  $\pi(s,a) \leftarrow \begin{cases}1, & \text{if } a = \text{argmax}_{a'}\left(r(s,a')+C(s,a')\right) \\ 0, & \text{otherwise}\end{cases}$   $\forall s,a$

---

Using $C$ function, we can formulate a Policy Iteration algorithm as shown in Algorithm 1, which is guaranteed to converge to an optimal policy (Theorem 1), similar to the case of Policy Iteration with $Q$ or $V$ function [16]. Other properties of $C$ function and proofs are presented in Appendix B.

## 4.3 Continuous-control using Actor Residual Critic

We can easily extend the policy iteration algorithm with $C$ function (Algorithm 1) for continuous-control tasks using function approximators instead of discrete $C$ values and a discrete policy (similar to the case of $Q$ function [16]). We call any RL algorithm that uses a policy, $\pi$ and a residual critic, $C$ function as an Actor Residual Critic (ARC) algorithm. Using the specific details of different existing Actor Critic algorithms, we can formulate analogous ARC algorithms. For example, using a deterministic policy and target network as in [17] we can get ARC-DDPG. Using double C networks (instead of Q networks) and delayed updates as in [18] we can get ARC-TD3. Using entropy regularization and reparameterization trick as in [3] we can get ARC-SAC or SARC (Soft Actor Residual Critic).

## 4.4 ARC aided Adversarial Imitation Learning

To incorporate ARC in any Adversarial Imitation Learning algorithm, we simply replace the Actor Critic RL algorithm with an ARC RL algorithm without altering anything else in the pipeline. For example, we can replace SAC [3] with SARC to get SARC-AIL as shown in Algorithm 2. Implementation-wise this is extremely simple and doesn't require any additional functional parts in the algorithm. The same neural network that approximated $Q$ function can be now be used to approximate $C$ function.

---

**Algorithm 2:** SARC-AIL: Soft Actor Residual Critic Adversarial Imitation Learning

---

**Intialization**: Environment (env), Discriminator parameters $\psi$, Policy parameters $\theta$, $C$-function parameters $\phi_1$, $\phi_2$, dataset of expert demonstrations $\mathcal{D}^{\text{exp}}$, replay buffer $\mathcal{D}$, Target parameters $\phi_{\text{targ}1} \leftarrow \phi_1, \phi_{\text{targ}2} \leftarrow \phi_2$, Entropy regularization coefficient $\alpha$;

**while** *Max no. of environment interactions is not reached* **do**

  $a \sim \pi_\theta(.|s)$;

  $s', r, d = $ env.step(a);   $d = 1$ if $s'$ is terminal state, 0 otherwise

  Store $(s, a, s', d)$ in replay buffer $\mathcal{D}$;

  **if** *Update interval reached* **then**

    **for** *no. of update steps* **do**

      Sample batch $B = (s, a, s', d) \sim \mathcal{D}$;

      Sample batch of expert demonstrations $B^{\text{exp}} = (s, a) \sim \mathcal{D}^{\text{exp}}$;

      Update Discriminator parameters ($\psi$) with gradient ascent.

$$\nabla_\psi \left\{ \sum_{(s,a)\in B^{\text{exp}}}[\log D_\psi(s,a)] + \sum_{(s,a,s',d)\in B}[\log(1 - D_\psi(s,a))] \right\};$$

      Compute $C$ targets $\forall (s, a, s', d) \in B$

$$y(s,a,d) = \gamma \left( r_\psi(s', \tilde{a}') + \min_{i=1,2} C_{\phi_{\text{targ}i}}(s', \tilde{a}') - \alpha \log \pi_\theta(\tilde{a}'|s') \right), \quad \tilde{a}' \sim$$
$$\pi_\theta(.|s'), r_\psi(s', \tilde{a}') = h(D_\psi(s,', \tilde{a}'))$$

      Update C-functions parameters ($\phi_1, \phi_2$) with gradient descent.

$$\nabla_{\phi_i} \frac{1}{|B|} \sum_{(s,a,s',d)\in B} (C_{\phi_i}(s,a) - y(s,a,d))^2, \quad \text{for } i = 1, 2$$

      Update policy parameters ($\theta$) with gradient ascent.

$$\nabla_\theta \frac{1}{|B|} \sum_{s\in B} \left( r_\psi(s, \tilde{a}) + \min_{i=1,2} C_{\phi_i}(s, \tilde{a}) - \alpha \log \pi_\theta(\tilde{a}|s) \right), \quad \tilde{a} \sim$$
$$\pi_\theta(.|s), r_\psi(s, \tilde{a}) = h(D_\psi(s, \tilde{a}))$$

      Update target networks.

$$\phi_{\text{targ}i} \leftarrow \zeta \phi_{\text{targ}i} + (1 - \zeta)\phi_i, \quad \text{for } i = 1, 2; \quad \zeta \text{ controls polyak averaging}$$

---

## 4.5 Why choose ARC over Actor-Critic in Adversarial Imitation Learning?

The advantage of using an ARC algorithm over an Actor-Critic (AC) algorithm is that we can leverage the exact gradient of the reward. Standard AC algorithms use $Q_\phi$ to approximate the immediate reward + future return and then compute the gradient of the policy parameters through the $Q_\phi$ function (12). This is an approximate gradient with no bound on the error in gradient, since the $Q_\phi$ function is an estimated value, Appendix A.1. On the other hand, ARC algorithms segregate the immediate reward (which is known in Adversarial Imitation Learning) from the future return (which needs to be estimated). ARC algorithms then compute the gradient of policy parameters through the immediate reward (which is exact) and the $C$ function (which is approximate) separately (13).

$$\text{Standard AC} \quad \mathbb{E}_{s\sim\mathcal{D}}\left[ \nabla_\theta Q_\phi(s,a) \right], \qquad\qquad a \sim \pi_\theta(.|s) \qquad (12)$$

$$\text{ARC (Our)} \quad \mathbb{E}_{s\sim\mathcal{D}}\left[ \nabla_\theta r(s,a) + \nabla_\theta C_\phi(s,a) \right], \qquad a \sim \pi_\theta(.|s) \qquad (13)$$

In Appendix A.2, we derive the conditions under which ARC is likely to outperform AC by performing a (Signal to Noise Ratio) SNR analysis similar to [34]. Intuitively, favourable conditions

for ARC are (i) Error in gradient due to function approximation being similar or smaller for $C$ as compared to $Q$ (ii) the gradient of the immediate reward not having a high negative correlation with the gradient of $C$ ($\mathbb{E}\left[\nabla_a r(s,a)\nabla_a C(s,a)\right]$ is not highly negative). Under these conditions, ARC would produce a higher $SNR$ estimate of the gradient to train the policy. We believe that AIL is likely to present favourable conditions for ARC since the reward is shaped.

ARC would under-perform AC if the error in gradient due to function approximation of $C$ network is significantly higher than that of $Q$ network. In the general RL setting, immediate reward might be misleading (i.e. $\mathbb{E}\left[\nabla_a r(s,a)\nabla_a C(s,a)\right]$ might be negative) which might hurt the performance of ARC. However, we propose using ARC for AIL where the adversary reward measures how closely the agent imitates the expert. In AIL, the adversary reward is dense/shaped making ARC likely to be useful in this scenario, as experimentally verified in the following section.

## 5 Results

In Theorem 1, we proved that Policy Iteration with $C$ function converges to an optimal policy. In Fig. 2, we experimentally validate this on an example grid world. The complete details are presented in Appendix E.1. In the following sections (5.2, 5.3 and 5.4) we show the effectiveness of ARC aided AIL in Mujoco continuous-control tasks, and simulated and real robotic manipulation tasks. In Appendix D.2, we experimentally illustrate that ARC produces more accurate gradients than AC using a simple 1D driving environment. The results are discussed in more detail in Appendix F.

### 5.1 Policy Iteration on a Grid World

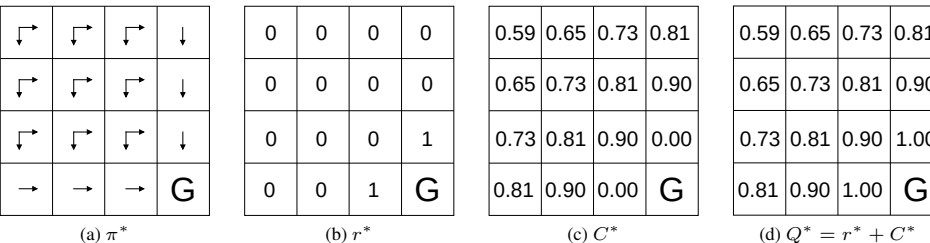

(a) $\pi^*$  (b) $r^*$  (c) $C^*$  (d) $Q^* = r^* + C^*$

Figure 2: On a Grid World, the results of running two Policy Iteration (PI) algorithms - PI with $C$ function (Algorithm 1) and the standard PI with $Q$ function (Appendix C.1 Algorithm 3). Both algorithms converge in 7 policy improvement steps to the same optimal policy $\pi^*$ as shown in a. The optimal policy gets the immediate reward shown shown b. The $C$ values c at the convergence of PI with $C$ function and the $Q$ values d at the convergence of PI with $Q$ function are consistent with their relation $Q^* = r^* + C^*$ (11). Details are in E.1.

### 5.2 Imitation Learning in Mujoco continuous-control tasks

We used 4 Mujoco continuous-control environments from OpenAI Gym [35], as shown in Fig. 3. Expert trajectories were obtained by training a policy with SAC [3]. We evaluated the benefit of using ARC with two popular Adversarial Imitation Learning (AIL) algorithms, $f$-MAX-RKL [12] and GAIL [10]. For each of these algorithms, we evaluated the performance of standard AIL algorithms ($f$-MAX-RKL, GAIL), ARC aided AIL algorithms (ARC-$f$-MAX-RKL, ARC-GAIL) and Naive-Diff algorithm described in Section 3 (Naive-Diff-$f$-MAX-RKL, Naive-Diff-GAIL). We also evaluated the performance of Behavior Cloning (BC). For standard AIL algorithms (GAIL and $f$-MAX-RKL) and BC, we used the implementation of [28]. Further experimental details are presented in Appendix E.

### 5.3 Imitation Learning in robotic manipulation tasks

We used simplified 2D versions of FetchReach (Fig. 5a) and FetchPush (Fig. 5b) robotic manipulation tasks from OpenAI Gym [35] which have a simulated Fetch robot, [36]. In the FetchReach task, the robot needs to take it's end-effector to the goal (virtual red sphere) as quickly as possible. In the FetchPush task, the robot's needs to push the block to the goal as quickly as possible. We used hand-coded proportional controller to generate expert trajectories for these tasks. Further details are presented in Appendix E.3.

Fig. 4 shows the training plots and Table 2 shows the final performance of the different algorithms. Across all environments and across both the AIL algorithms, incorporating ARC shows consistent improvement over standard AIL algorithms (Table 2). BC suffers from distribution shift at test time [19, 10] and performs very poorly. As we predicted in Section 3, Naive-Diff algorithms don't perform well as naively using autodiff doesn't compute the gradients correctly.

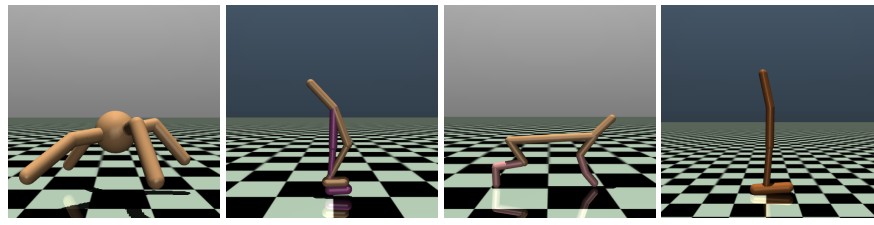

(a) Ant-v2      (b) Walker-v2      (c) HalfCheetah-v2      (d) Hopper-v2

Figure 3: OpenAI Gym's [35] Mujoco continuous-control environments used for evaluation.

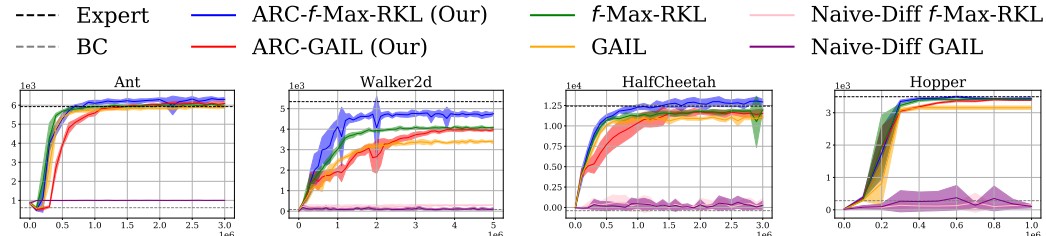

Figure 4: Episode return versus number of environment interaction steps for different Imitation Learning algorithms on Mujoco continuous-control environments.

| Method | Ant | Walker2d | HalfCheetah | Hopper |
|---|---|---|---|---|
| Expert return | 5926.18 ± 124.56 | 5344.21 ± 84.45 | 12427.49 ± 486.38 | 3592.63 ± 19.21 |
| ARC-$f$-Max-RKL (Our) | **6306.25** ± 95.91 | **4753.63** ± 88.89 | **12930.51** ± 340.02 | **3433.45** ± 49.48 |
| $f$-Max-RKL | 5949.81 ± 98.75 | 4069.14 ± 52.14 | 11970.47 ± 145.65 | 3417.29 ± 19.8 |
| Naive-Diff $f$-Max-RKL | 998.27 ± 3.63 | 294.36 ± 31.38 | 357.05 ± 732.39 | 154.57 ± 34.7 |
| ARC-GAIL (Our) | **6090.19** ± 99.72 | **3971.25** ± 70.11 | **11527.76** ± 537.13 | **3392.45** ± 10.32 |
| GAIL | 5907.98 ± 44.12 | 3373.26 ± 98.18 | 11075.31 ± 255.69 | 3153.84 ± 53.61 |
| Naive-Diff GAIL | 998.17 ± 2.22 | 99.26 ± 76.11 | 277.12 ± 523.77 | 105.3 ± 48.01 |
| BC | 615.71 ± 109.9 | 81.04 ± 119.68 | -392.78 ± 74.12 | 282.44 ± 110.7 |

Table 2: Policy return on Mujoco environments using different Imitation Learning algorithms. Each algorithm is run with 10 random seeds. Each seed is evaluated for 20 episodes.

Fig. 6a shows the training plots and Table 3 under the heading 'Simulation' shows the final performance of the different algorithms. In both the FetchReach and FetchPush tasks, ARC aided AIL algorithms consistently outperformed the standard AIL algorithms. Fig. 6b shows the magnitude of the $2^{nd}$ action dimension vs. time-step in one episode for different algorithms. The expert initially executed large actions when the end-effector/block was far away from the goal. As the end-effector/block approached the goal, the expert executed small actions. ARC aided AIL algorithms (ARC-$f$-Max-RKL and ARC-GAIL) showed a similar trend while standard AIL algorithms ($f$-Max-RKL and GAIL) learnt a nearly constant action. Thus, ARC aided AIL algorithms were able to better imitate the expert than standard AIL algorithms.

### 5.4   Sim-to-real transfer of robotic manipulation policies

For testing the sim-to-real transfer of the different trained AIL manipulation policies, we setup JacoReach (Fig. 5c) and JacoPush (Fig. 5d) tasks with a Kinova Jaco Gen 2 arm, similar to the FetchReach and FetchPush tasks in the previous section. The details are presented in Appendix E.4.

Table 3 under the heading 'Real Robot' shows the performance of the different AIL algorithms in the real robotic manipulation tasks. The real robot evaluations showed a similar trend as in the simulated tasks. ARC aided AIL consistently outperformed the standard AIL algorithms. Appendix D Fig. 9 visualizes the policies in the JacoPush task showing that ARC aided AIL algorithms were able to push the block closer to the goal as compared to the standard AIL algorithms. Supplementary slide shows videos of the same. Since we didn't tune hyper-parameters for these tasks (both our methods and the baselines, details in Appendix E.3), it is likely that the performances would improve with further parameter tuning. Without fine-tuning hyper-parameters for these tasks, ARC algorithms showed higher performance than the baselines. This shows that ARC algorithms are parameter robust and applicable to real robot tasks without much fine tuning.

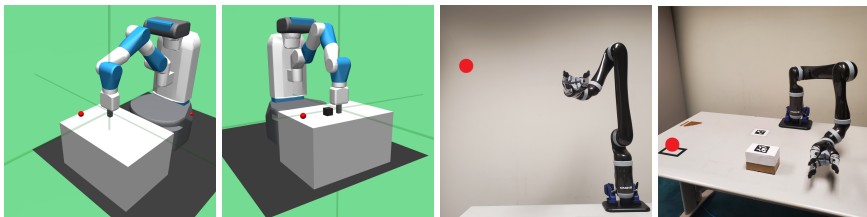

|(a) FetchReach|(b) FetchPush|(c) JacoReach|(d) JacoPush|

Figure 5: Simulated and real robotic manipulation tasks used for evaluation. Simplified 2D versions of the FetchReach a and FetchPush b tasks from OpenAI Gym, [35] with a Fetch robot, [36]. Corresponding JacoReach c and JacoPush d tasks with a real Kinova Jaco Gen 2 arm, [37].

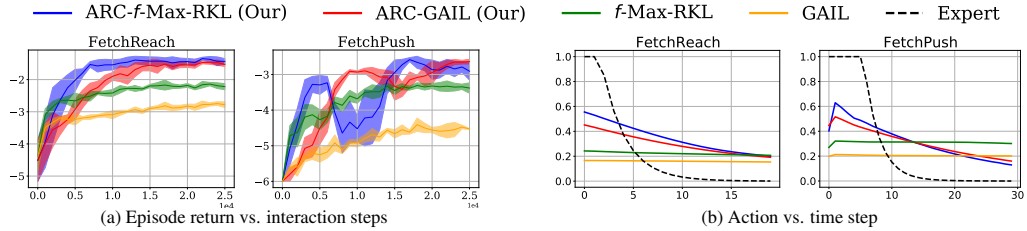

|(a) Episode return vs. interaction steps|(b) Action vs. time step|

Figure 6: a Episode return vs. number of environment interaction steps for different Adversarial Imitation Learning algorithms on FetchPush and FetchReach tasks. b Magnitude of the $2^{nd}$ action dimension versus time step in a single episode for different algorithms.

| | Simulation | | Real Robot | |
| Method | FetchReach | FetchPush | JacoReach | JacoPush |
|---|---|---|---|---|
| Expert return | $-0.58 \pm 0$ | $-1.18 \pm 0.04$ | $-0.14 \pm 0.01$ | $-0.77 \pm 0.01$ |
| ARC-$f$-Max-RKL (Our) | $\mathbf{-1.43} \pm 0.08$ | $\mathbf{-2.91} \pm 0.25$ | $\mathbf{-0.38} \pm 0.02$ | $\mathbf{-1.25} \pm 0.06$ |
| $f$-Max-RKL | $-2.22 \pm 0.09$ | $-3.38 \pm 0.15$ | $-0.8 \pm 0.05$ | $-2.03 \pm 0.06$ |
| ARC-GAIL (Our) | $\mathbf{-1.53} \pm 0.06$ | $\mathbf{-2.64} \pm 0.07$ | $\mathbf{-0.46} \pm 0.01$ | $\mathbf{-1.56} \pm 0.08$ |
| GAIL | $-2.78 \pm 0.09$ | $-4.53 \pm 0.01$ | $-1.05 \pm 0.06$ | $-2.35 \pm 0.06$ |

Table 3: Policy return on simulated (FetchReach, FetchPush) and real (JacoReach, JacoPush) robotic manipulation tasks using different AIL algorithms. The reward at each time step is negative distance between end-effector & goal for reach tasks and block & goal for push tasks. The reward in the real and simulated tasks are on different scales due to implementation details described in Appendix E.4.

## 6 Limitations

Three main limitations in our work are: (1) While many AIL algorithms can be trained using expert 'states' only, ARC-AIL can only be trained with 'state-action' $(s, a)$ pairs. There are several scenarios where obtaining $(s, a)$ pairs is challenging (e.g. kinesthetic teaching). In such scenarios, ARC is not directly applicable. People often use tricks to mitigate this issue and using $(s, a)$ pairs to train a policy is a popular choice [38, 39, 40, 41, 42]. (2) ARC-AIL can only work with continuous action space. Most real world robotic tasks have or can be modified to have a continuous action space. (3) We haven't explored how the agent-adversary interaction in AIL affects the accuracy of the reward gradient and leave that for future work.

## 7 Conclusion

We highlighted that the reward in popular Adversarial Imitation Learning (AIL) algorithms are differentiable but this property has not been leveraged by existing model-free RL algorithms to train a policy. Further, they are usually shaped. We also showed that naively differentiating the policy through this reward function does not perform well. To solve this issue, we proposed a class of Actor Residual Critic (ARC) RL algorithms that use a $C$ function as an alternative to standard Actor Critic (AC) algorithms which use a $Q$ function. An ARC algorithm can replace the AC algorithm in any existing AIL algorithm. We formally proved that Policy Iteration using $C$ function converges to an optimum policy in tabular environments. For continuous-control tasks, using ARC can compute the exact gradient of the policy through the reward function which helps improve the performance of the AIL algorithms in simulated continuous-control and simulated & real robotic manipulation tasks. Future work can explore the applicability of ARC algorithm to other scenarios which have a differentiable reward function.

**Acknowledgments**

We are thankful to Swaminathan Gurumurthy and Tejus Gupta for several insightful discussions on the idea. We are also thankful to Rohit Jena, Advait Gadhikar for their feedback on the manuscript and Dana Hughes, Sushmita Das for their support with some logistics of the project.

Finally, we are thankful to the reviewers for their constructive feedback through the rebuttal period, which we believe helped strengthen the paper.

This work has been supported by the following grants: Darpa HR001120C0036, AFRL/AFOSR FA9550-18-1-0251 and ARL W911NF-19-2-0146.

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
