# OpenReview forum: "ARC - Actor Residual Critic for Adversarial Imitation Learning"
_robot-learning.org/CoRL/2022/Conference — CoRL 2022 Poster_

### Official Review · Reviewer_XgGH · 2022-07-26

**Originality:** Good
**Technical Quality:** Good
**Clarity Of Presentation:** Good
**Impact:** 3

**Recommendation:**

Weak Accept: I recommend accepting the paper, but will not argue for my recommendation if the majority of other reviewers have a different opinion.

**Summary:**

This paper proposes a method for updating policies using the gradient of the reward function in adversarial imitation learning. The Q function is represented as the sum of the immediate reward and the C function approximates the sum of later rewards, and the gradient of this Q function is used to update policies. Simulations verified the proposed method to outperform standard adversarial imitation learning.

**Issues:**

See weaknesses above.

**Quality Of The Limitations Section:**

Limitations are addressed clearly

**Reviewer Expertise:**

3: The reviewer is fairly confident that the evaluation is correct

**Robotics Focus:**

Relevant but unlikely to deploy to hardware in near future

**Strengths And Weaknesses:**

Strengths:
* Proposal for adversarial imitation learning using gradients of an immediate reward in policy updating with model-free RL
* Basic validation with simple toy tasks
* The proposed method's performance was investigated in various simulation tasks

Weaknesses:
* The proposal in this study is not a method for solving a real robot task problem. It is a proposal for general adversarial imitation learning.
* There is not enough explanation as to why the proposed method improves performance. Since the implementation simply considers the gradient of the immediate reward, a little more explanation of the reason is needed.
* I do not understand why GAIL's performance is too low in the Fetch and Jaco tasks in Sections 5.3 and 5.4. I think these tasks are not very difficult and GAIL can learn if the parameter settings are appropriate. I checked the readme of the code to confirm the GAIL parameters, but there was no description of config description for these reaching tasks.

**Summary Of Recommendation:**

The proposed method is applicable to a wide range of applications because it improves general adversarial imitation learning. The implementation of the proposed method is simple. The appendix is extensive.

---

> ### Author Response · Authors · 2022-08-18
> **Response**
>
> Thank you for your valuable feedback. Here are our responses to the weaknesses:
>
> 1. **No real robot**: We have shown our results on a **real robot**. We show a **real Jaco robot arm** performing 2 tasks - reaching a target goal location and pushing an object to a goal location. Kindly refer to Section 5.4, Appendix D1 and supplementary slides with embedded videos.
>
> 1. **More explanation on why performance improves**: Both standard Actor Critic (AC) and proposed ARC estimate the gradient of the Q function to update the policy parameters. In ARC, we leverage the exact gradient of the immediate reward which helps produce a more accurate estimate of the Q gradient. This provides a better training signal. This is explained in more detail in Section 4.5, Appendix A2 and empirically verified in Appendix D2.
>
> 1. **Poor performance of GAIL in Fetch tasks**: We agree and have already mentioned this in the Appendix E3, 3rd paragraph. We reiterate that here:
> For each AIL algorithm, once every 20 environment steps, the discriminator and the policy were alternately trained for 10 iterations each. Each AIL algorithm was trained for 25,000 environment steps. All the other hyper-parameters were the same as those used with the Ant, Walker and Hopper Mujoco continuous-control environments (Section E2). We didn't perform any hyper-parameter tuning (for both our methods and the baselines) in these experiments and the results might improve with some hyper-parameter tuning.
>
> Kindly let us know if we can provide additional information.

---

> > ### Comment · Reviewer_XgGH · 2022-08-19
> > **Re: Response**
> >
> > Thanks for the reply. Since I am convinced about response 2, I will comment on responses 1 and 3.
> >
> > * Response 1: I evaluate that the policies learned in the simulations were applied to real robot tasks. I mean, I would like to point out that this study did not focus on the problems that occur when applying AIL to real robot tasks. For example, it is clearer to understand the worth of this study in the robot learning domain if you claim the following: if the proposed method is parameter robust as mentioned in response 3, then the proposed method is suitable for real robot tasks with high learning costs.
> >
> >
> > * Response 3: It is good that the proposed method is parameter robust. The unnaturally low performance of the comparison method appears suspicious, and it would be better to add the reason to Section 5.4.

---

> > > ### Author Response · Authors · 2022-08-19
> > > **Response**
> > >
> > > Thank you for the insightful feedback:
> > > * Response 1: Thank you for pointing this out. "Parameter robustness" looks like a positive point for real world robotics that we missed adding to the paper.
> > > * Response 3: For the baselines, we used the implementation of [28] from CoRL 2020. Parameters were inspired by the same. The baselines performed quite well in the mujoco environments. These are evidences suggesting that the baseline implementations are correct. The reason for low performance of baselines in robotic tasks, as you pointed out, might be that ARC methods seem to be more parameter robust (since we didn't fine tune both ARC methods and baselines for the robotic tasks and yet ARC methods performed well on the robotic tasks).
> > >
> > >  Upon your recommendation we have added these points to Section 5.4.
> > >
> > > **Kindly let us know if we have sufficiently addressed your concerns or if you have any additional comments.**

---

> > > > ### Author Response · Authors · 2022-08-20
> > > > **Awaiting a response**
> > > >
> > > > Dear Reviewer,
> > > >
> > > > We are awaiting your response. Kindly let us know if we have addressed your concerns or if you have any additional comments.
> > > >
> > > > Looking forward to hearing from you.

---

> > > > > ### Comment · Reviewer_XgGH · 2022-08-21
> > > > > **Re: Awaiting a response**
> > > > >
> > > > > I checked the additional explanation and there are no problems.

---

> > > > > > ### Author Response · Authors · 2022-08-21
> > > > > > **Thank you**
> > > > > >
> > > > > > We would like to thank you for the positive feedback.

---

### Official Review · Reviewer_AjRj · 2022-07-29

**Originality:** Good
**Technical Quality:** Good
**Clarity Of Presentation:** Good
**Impact:** 3

**Recommendation:**

Weak Accept: I recommend accepting the paper, but will not argue for my recommendation if the majority of other reviewers have a different opinion.

**Summary:**

In adversarial imitation learning (e.g. GAIL), the reward function is expressed in terms of the discriminator output and thus differentiable. When an actor-critic method is used for optimizing this reward, the analytic and differentiable form of this reward can be exploited by expressing the Q function as $Q^{\pi}(s, a) = r(s, a) + C^{\pi}(s, a)$ and only learning the future return $C(s, a)$ during policy evaluation. The motivation behind this parameterization is that during policy improvement, the known gradient of $r(s,a)$ is used, while approximation errors only affect the gradient of the future return. The paper presents a theorem to show convergence of policy iteration on the C-function, a theoretical justification arguing that the signal-to-noise ratio of the gradient is larger when learning the C-function, and experimental results on Mujoco and Sim-to-Real transfer to a Fetch robot.


**Issues:**

- The paper should provide at least empirical results on when the decomposition helps and **when it does not help**. It would be for example interesting to perform and discuss an experiment that is similar to the car-experiment but with a reward function that has action costs and a final state reward.
- It needs to be clarified if the proposed method differs in any way from using a Q-network parameterized as $Q_{\theta}(s,a) = r(s,a) + C_{\theta}(s,a)$. If that is the case, it probably does not make sense to introduce a new policy iteration (Theorem 1) could be dropped.
- The limitation section should be improved.


**Quality Of The Limitations Section:**

Additional details required

**Reviewer Expertise:**

5: The reviewer is absolutely certain that the evaluation is correct and very familiar with the relevant literature

**Robotics Focus:**

Sufficient demonstration on hardware

**Strengths And Weaknesses:**

The paper is well-written and proposes a simple modification to existing AIL method that can be easily implemented for slightly improving the performance.

However, the quality of submission suffers significantly from overselling, which resulted in 1) (a quite blatant) error, 2) overcomplicating the approach, 3) blurring the contributions:
1) (_Error_:) The paper aims to prove that learning the C function is better (lower variance) than learning the Q function, in general. I argue, that this is actually not the case, as we can easily construct counter-examples, where the gradient of the immediate reward function is misleading, for example when the agent needs to perform several bad-reward-actions to finally reach a highly-desirable final state. I conjecture, that in such settings, learning the C-Function would rather hurt the performance than improve it. In the misguided effort to prove the general advantage of the proposed decomposition in A.2, the paper wrongfully assumes that the SNR of a combined signal corresponds to the sum of the individual signal-to-noise ratios.

2) (_Overcomplication_:) The paper contains a theorem for the sake of containing a theorem. I can not see, why we need a special Bellman-iteration for iterating over $C$. For all, I can tell, we can simply assume the proposed modification as imposing a certain structure on the Q-function ($Q(s,a) = r(s,a) + C(s,a)$) and performing standard updates on the Q-function. This simplicity is a strong selling point! We can modifiy existing Actor-Critic-AIL methods, by adding one line of code in the Q-network. Why do the authors scare people off, by introducing a "novel" policy iteration?

3) (_Blurring the Contribution_:) In my opinion, the main insight is not that we can use knowledge of the reward gradient during policy iteration. I would argue that the reason that people did not propose this in a reinforcement learning setting (where often enough, we also have differentiable reward functions), is that learning the C-function is not beneficial per se. The intriguing bit about the current submission is, that *in the adversarial imitation learning setting, learning the C-function might indeed be beneficial because the reward function is typically shaped ([11])*. Interestingly, also the 1-D example in D.2 which demonstrates lower approximation errors when using the $r$-$C$ decomposition, uses a shaped reward function that is given by the likelihood of the expert polciy (with Gaussian action noise). Based on my interpretation of the mechanics of the proposed method, reward-shaping is a key factor for the benefits of the proposed method that is not even touched in current submission.

The limitations are not well discussed:
- I think that the paper should mention that it does not provide insights on when the C-decomposition help, and when it doesn't. The paper does not provide a solid theoretical understanding on the effects of the decomposition on the sample efficiency. Providing formal results based on certain assumptions would be an important future work. I could, for example, imaging that decreased gradient variance could be formally proven when assuming shaped rewards of the form $r(s, a) = \log \pi^{\star}(a|s)$

- The limitation section mentions the problem that actions must be observed, but is quick to dismiss this with the assertion that actions can be typically provided in robotics. I do not think that this is a just assessment. Arguably the most relevant scenarios for providing demonstrations are 1) observing human demonstrations and 2) kinesthetic teaching. For human demonstrations, it is already difficult to map the states to the robot (e.g. by mapping marker positions), but obtaining control commands is hardly possible. For kinesthetic teaching we can measure the joint positions of the robot during demonstrations, but even if the robot is equipped with joint torque sensors, accurately distinguishing external and internal torques is typically not possible. Demonstrating by teleoperation, and demonstrating by running a given controller are to my knowledge the most realistic settings for imitation learning with action, but they are, arguably, of less practical relevant compared to the aforementioned scenarios.



**Summary Of Recommendation:**

While I believe that the paper touches an interesting idea, the current presentation is too misleading and seems to contain a major error in the theoretical motivation (SNR).

---

> ### Author Response · Authors · 2022-08-18
> **Response**
>
> Thank you for your insightful feedback. Kindly
>
> 1. Take a look at our response to each of your concerns in separate comments.
> 1. Refer to the **Revision 1** of the paper: https://openreview.net/forum?id=V3Mjpi4kzdn&noteId=vx6_73pWIV
>
> Kindly let us know if we missed addressing any of your concerns.

---

> ### Author Response · Authors · 2022-08-18
> **Response to (1) *Error***
>
> **(A) Response to counter example**
>
> There are 2 rewards:
> * environment reward (i.e. the standard MDP reward)
> * adversary reward (reward obtained from adversary in AIL)
>
> It is possible that the immediate *environment reward* is misleading. However, the *adversary reward* is a measure of closeness between agent and expert actions. It naturally is never misleading (assuming a reasonable adversary). If we have an initial bad action that the expert takes to obtain a high reward later on, then the initial bad action will have a corresponding high *adversary reward*. This is because the expert took that action and the agent would get high *adversary reward* for taking the same action.
>
>
> **(B) Response to wrongfully assumes that the SNR is additive**
>
> We humbly point out that we didn’t assume that SNR is additive. In the derivation in Appendix A2, we added signal strengths and not SNR’s directly.
>
> Here are the detailed steps in the relevant part of the derivation:
>
> 1. The signal strength from $\nabla_a r(s,a)$ is 1 (noise = 0, total strength = 1).
> 1. $\nabla_a \hat{C}(s,a)$ has signal to noise ratio due to function approximation equal to $\texttt{snr}$. Thus in $\nabla_a \hat{C}(s,a)$, the signal strength is $\frac{\texttt{snr}}{1+\texttt{snr}}$ and the noise strength is $\frac{1}{1+\texttt{snr}}$ (total strength = 1).
> 1. The net signal strength in $\nabla_a r(s,a) + \nabla_a \hat{C}(s,a)$ is the addition of the individual signal strengths of $\nabla_a r(s,a)$ and $\nabla_a \hat{C}(s,a)$. Thus the net signal strength is $1+\frac{\texttt{snr}}{1+\texttt{snr}}$, assuming equal weightage from the gradients of $r$ and $\hat{C}$.
> 1. The net noise strength in $\nabla_a r(s,a) + \nabla_a \hat{C}(s,a)$ is the addition of the individual noise strengths. Thus the net noise strength is $0 + \frac{1}{1+\texttt{snr}} = \frac{1}{1+\texttt{snr}}$.
> 1. Thus the effective signal to noise ratio is $\frac{\text{net signal strength}}{\text{net noise strength}}$. This is equal to $\frac{1 + \frac{\texttt{snr}}{1+\texttt{snr}}}{\frac{1}{1+\texttt{snr}}} = 2\texttt{snr} + 1 > \texttt{snr}$
>
> **Kindly let us know if this derivation looks correct or not and if you'd like us to provide further clarification.**

---

> > ### Comment · Reviewer_AjRj · 2022-08-18
> > **Questions regarding SNR derivations**
> >
> > Thanks, for the reply. Unfortunately I can still not follow your argumentation.
> >
> > Why is the signal strength of $\nabla_a \hat{C}(s,a) = \frac{\text{snr}}{1 + \text{snr}}$? (Why is the total signal strength 1?)

---

> > > ### Author Response · Authors · 2022-08-18
> > > **Derivation of signal strength**
> > >
> > > Thank you for the quick reply and the follow-up question.
> > >
> > > For simplification, assume total strength of $\nabla_a \hat{C}(s,a)$ is 1 (we could assume any magnitude and it’d scale both the signal and noise strengths by the same factor):
> > >
> > > $$signal \textunderscore strength + noise \textunderscore strength = 1 \quad \quad (i)$$
> > >
> > > By definition of SNR:
> > >
> > > $$\begin{aligned}
> > > \texttt{snr} &= \frac{signal \textunderscore strength}{noise \textunderscore strength} \newline
> > > \implies noise \textunderscore strength &= \frac{signal  \textunderscore strength}{\texttt{snr}} \quad \quad (ii)
> > > \end{aligned}$$
> > >
> > > Substituting (ii) in (i):
> > > $$\begin{aligned}
> > > signal  \textunderscore strength + \frac{signal  \textunderscore strength}{\texttt{snr}} &= 1 \newline
> > > \implies signal  \textunderscore strength \left(1 + \frac{1}{\texttt{snr}}\right) &= 1 \newline
> > > \implies signal  \textunderscore strength \left(\frac{\texttt{snr} + 1}{\texttt{snr}}\right) &= 1 \newline
> > > \therefore signal  \textunderscore strength &= \frac{\texttt{snr}}{1+\texttt{snr}}
> > > \end{aligned}$$
> > >
> > > **Kindly let us know if this looks correct or not and if we can provide additional details.**

---

> > > > ### Comment · Reviewer_AjRj · 2022-08-19
> > > > **Re: Derivation of signal strength**
> > > >
> > > > in 1. you assumed that the signal strength of $\nabla r$ equals 1. It seems like you can not further assume wlog that the total strength of $\nabla C$ equals 1, since scaling one value would also change the other quantity.
> > > >
> > > > I also don't see how the scalar SNR is computed from the vector, in particular when we want to draw meaningful conclusions wrt the gradient variance.

---

> > > > > ### Author Response · Authors · 2022-08-19
> > > > > **Response**
> > > > >
> > > > > Thank you for the insightful follow-up.
> > > > >
> > > > > **PART 1: General Derivation**
> > > > >
> > > > > We made the simplification to convey the basic idea. Upon your recommendation, we show in the following the general derivation **without assuming** that the strength of $\nabla_a C(s,a)$ is 1.
> > > > >
> > > > > 1. The signal strength from $\nabla_a r(s,a)$ is 1 (noise = 0, total strength = 1).
> > > > > 1. Let the total strength of $\nabla_a \hat{C}(s,a)$ be $\lambda$. $\nabla_a \hat{C}(s,a)$ has signal to noise ratio due to function approximation equal to $\texttt{snr}$. Thus in $\nabla_a \hat{C}(s,a)$, the signal strength is $\frac{\lambda \texttt{snr}}{1+\texttt{snr}}$ and the noise strength is $\frac{\lambda}{1+\texttt{snr}}$ (total strength = $\lambda$).
> > > > > 1. The net signal strength in $\nabla_a r(s,a) + \nabla_a \hat{C}(s,a)$ is the addition of the individual signal strengths of $\nabla_a r(s,a)$ and $\nabla_a \hat{C}(s,a)$. Thus the net signal strength is $1+\frac{\lambda \texttt{snr}}{1+\texttt{snr}}$.
> > > > > 1. The net noise strength in $\nabla_a r(s,a) + \nabla_a \hat{C}(s,a)$ is the addition of the individual noise strengths. Thus the net noise strength is $0 + \frac{\lambda}{1+\texttt{snr}} = \frac{\lambda}{1+\texttt{snr}}$.
> > > > > 1. Thus the effective signal to noise ratio is $\frac{\text{net signal strength}}{\text{net noise strength}}$. This is equal to $\frac{1 + \frac{\lambda \texttt{snr}}{1+\texttt{snr}}}{\frac{\lambda}{1+\texttt{snr}}} = \frac{1+\texttt{snr}+\lambda\texttt{snr}}{\lambda} = \frac{1+\texttt{snr}}{\lambda} + \frac{\lambda \texttt{snr}}{\lambda}= \frac{1+\texttt{snr}}{\lambda} + \texttt{snr}> \texttt{snr}$
> > > > >
> > > > > **Kindly let us know if this derivation looks correct or not and if you'd like us to provide further clarification.**
> > > > >
> > > > > **PART 2: The case for vectors**
> > > > >
> > > > > For a scalar, we have attempted to show that SNR improves. For a vector, we can treat each dimension of the vector as a scalar. Then the derivation holds true for each dimension of the vector separately. In other words, there is SNR improvement in each dimension of the vector individually. Intuitively, this is expected to lead to improvement in the overall vector signal.
> > > > >
> > > > > **Kindly let us know if you have additional comments.**

---

> > > > > > ### Comment · Reviewer_AjRj · 2022-08-19
> > > > > > **Regarding SNR "Proof"**
> > > > > >
> > > > > > Using a dimension of the gradient as a signal does not make sense, as it is in general not positive. Adding the "reward-gradient-signal" may in general very well decrease the total signal strength. The derivations essentially proves that $\nabla r + \nabla C > \nabla C$ based on the assumption $\nabla r = 1$, which you can not make wlog.
> > > > > >
> > > > > > I also don't see the point of comparing the snr of $\nabla r+C$ to the snr of $\nabla C$, and the assumption that "snr(Q) = snr(C)" is very strong and little useful. Furthermore using SNR as a proxy for gradient variance is not well motivated.
> > > > > >
> > > > > > I am still convinced that the paper seriously suffers from the fact that it tries to present theoretical proof for a general statement that simply does not hold. I would be fine with purely empirical and intuitive justification that the approach helps for AIL.
> > > > > >  Theoretical justification based on the assumption that the immediate reward is strongly correlated with the policy (which might be reasonable in the AIL setting) would of course be even better but it might be deferred to future work. However, misguiding derivations based on unreasonable assumptions drastically decrease the quality of the paper. I do not see any chance that you can turn A.2 into a convincing argument that "Decomposition in ARC leads to more accurate gradients", and I would strongly suggest to switch your strategy for the rebuttal and try to improve the submission, which still needs significant revisions to remove bold unsubstantiated statements.

---

> > > > > > > ### Author Response · Authors · 2022-08-20
> > > > > > > **Response**
> > > > > > >
> > > > > > > Thank you for the constructive feedback and a positive direction to improve the paper. Kindly refer to the updated paper: https://openreview.net/forum?id=V3Mjpi4kzdn&noteId=vx6_73pWIV
> > > > > > >
> > > > > > > 1. Upon your recommendation, we have reduced the boldness of our claims (replacing "would produce more accurate gradient" with "is likely to produce more accurate gradient in AIL").
> > > > > > >
> > > > > > > 1. We updated A2 (i) to have an argument based on intuition and (ii) a derivation on the **conditions when ARC helps** and the **conditions when it does not**. We have removed most of the assumptions except one assumption that the action space is 1D. Kindly look at the updated section A2.
> > > > > > >
> > > > > > > 1. At the end of A2, we have shown that there are 2 factors affecting the overall SNR:
> > > > > > >     * Relative magnitude of $\texttt{snr}_c$ and $\texttt{snr}_Q$ (SNR in $\nabla_a C(s,a)$ and $\nabla_a Q(s,a)$ respectively)
> > > > > > >     * Sign and magnitude of $S_{r,c} = \mathbb{E} [ \nabla_a r(s,a) \nabla_a C(s,a)]$ (i.e. whether gradient of reward and C are positively or negatively correlated)
> > > > > > >
> > > > > > > **Why is the decomposition likely to help in AIL?**
> > > > > > >
> > > > > > > ARC is likely to lead to higher SNR if $\texttt{snr}_c$ is not significantly smaller than $\texttt{snr}_Q$ and if $\mathbb{E} [ \nabla_a r(s,a) \nabla_a C(s,a)]$ is not highly negative.
> > > > > > >
> > > > > > > Both of these conditions are likely to hold in AIL because the reward is dense/shaped (as was rightly pointed out by you) and usually not misleading. Since $\texttt{snr}_c$, $\texttt{snr}_Q$ arise from function approximation of C and Q, and the reward is dense/shaped, it is likely that $\texttt{snr}_c$ is higher or at least similar in magnitude to $\texttt{snr}_Q$.  Similarly, since AIL reward is not misleading, $\mathbb{E} [ \nabla_a r(s,a) \nabla_a C(s,a)]$ is likely to be positive.
> > > > > > >
> > > > > > > **When would the decomposition hurt?**
> > > > > > >
> > > > > > > The decomposition is likely to hurt when the above conditions don't hold.
> > > > > > >
> > > > > > > To us, the results of the updated derivation seem sound and in line with the important factor pointed out by you that *the reward is AIL is shaped which helps ARC*. **Kindly let us know if the updated derivation in A2 looks good. If you are still not convinced, please let us know if we should completely remove it and base our arguments solely on intuition and experiments.**

---

> > > > > > > > ### Comment · Reviewer_AjRj · 2022-08-20
> > > > > > > > **Response**
> > > > > > > >
> > > > > > > > Thank you for your revisions. I did not have the time to check them yet, but I took a brief look at A2 and already have a question;
> > > > > > > >
> > > > > > > > You say $\nabla_a r(s,a)=E[\nabla_a (r(s, a)^2)]$.(and same for C): Why does this equality hold, what distributions are used for computing the expectation?

---

> > > > > > > > > ### Author Response · Authors · 2022-08-20
> > > > > > > > > **Response**
> > > > > > > > >
> > > > > > > > > Thank you again for the quick reply. We meant to say "Signal strength of $\nabla_a r(s,a)$" = $\mathbb{E}[(\nabla_a r(s,a))^2]$. This is from the definition of "signal strength"/"signal power".
> > > > > > > > >
> > > > > > > > > Reference: http://www.scholarpedia.org/article/Signal-to-noise_ratio. Please take a look at the Basics section, last paragraph. We use the 2nd definition of SNR.
> > > > > > > > >
> > > > > > > > > We also copy paste the text here:
> > > > > > > > >
> > > > > > > > > X=S+N , where both S and N are random variables.
> > > > > > > > > A random variable's power equals its mean-squared value: the signal power thus equals $𝖤[S^2]$ . Usually, the noise has zero mean, which makes its power equal to its variance. Thus, the SNR equals $𝖤[S^2]/σ^2_N$ .
> > > > > > > > >
> > > > > > > > > **Kindly let us know if anything else is unclear.**

---

> > > > > > > > > > ### Comment · Reviewer_AjRj · 2022-08-20
> > > > > > > > > > **Response**
> > > > > > > > > >
> > > > > > > > > > I see, please improve the notation to make clear that the first part of the text is supposed to be part of the equation.
> > > > > > > > > >
> > > > > > > > > > Regarding the new proof: I think it is better now, but I still think that the implications of the proof are oversold.
> > > > > > > > > >
> > > > > > > > > > * The statements that the sign of the reward gradient are "likely" to match the sign of the C-gradient seem bold. While I can imagine that there can be slight tendencies, I dont know of either theoretical or empirical evidence for such statement.
> > > > > > > > > >
> > > > > > > > > > * The following statement is misleading:
> > > > > > > > > > >  In other words, even if snrC is a fraction of snrQ, the net SNR due to decomposition is higher than that without decomposition.
> > > > > > > > > >
> > > > > > > > > > From what I understand, the statement needs to be reversed: Even if the gradients are positively correlated, snrC must be at least as large as a certain fraction of snrQ. The current statement sound like any fraction would be sufficient.
> > > > > > > > > >
> > > > > > > > > > * I still dont think that the SNR is a good proxy for the gradient variance. For example the assumption that the reward gradient is positively correlated with the true gradient of C (or Q) should already be enough to prove better convergence. So I don't see how the analysis of SNR is any meaningful.

---

> > > > > > > > > > > ### Author Response · Authors · 2022-08-20
> > > > > > > > > > > **Response**
> > > > > > > > > > >
> > > > > > > > > > > Thank you for taking the time to go through the proof and providing your insightful feedback.
> > > > > > > > > > >
> > > > > > > > > > > * First, kindly note that we do not strictly require the signs of the gradients to match. For both Case 1 ($S_{r,c}$>=0) and Case 2 ($-\frac{S_r S_c}{2}<S_{r,c}<0$): $\texttt{snr}_c$ needs to be at least as large as a certain fraction of $\texttt{snr}_Q$. Second, for AIL we can intuitively see why the 2 gradients are likely to be positively correlated:
> > > > > > > > > > >     * Let us say an expert follows a certain trajectory. As long as the agent follows the same trajectory, it would get high adversary reward. If the agent deviates from the trajectory, then it would get a low reward.
> > > > > > > > > > >     * If an action $a_1$ in state $s$ matches expert action, then it will have a high immediate reward $r(s,a_1)$. Since $a_1$ matches expert action, the agent would end up in next state $s'$ which is also a part of the trajectory, increasing the chances of following the trajectory correctly and getting higher reward in the future. Thus, $C(s,a_1)$ is also high since it approximates future reward.
> > > > > > > > > > >     * Similarly, if an action $a_2$ in state $s$ doesn't match expert action, then it will have a low immediate $r(s,a_2)$. Since $a_2$ doesn't match expert action, the agent is likely to deviate from the trajectory and go to the next state $s''$ (outside the trajectory). Once the agent goes outside the expert trajectory, it's chances of following the trajectory are lower. This means $C(s,a)$ is also low.
> > > > > > > > > > >     * Now lets say $a_1<a_2$ and they are close by values, i.e. $|a_2-a_1|$ tends to 0. If we compute the gradient $\nabla_a r(s,a)$ at $a_1$, then it will be negative since $a_2$ gives lower reward than $a_1$. Similarly,  $\nabla_a C(s,a)$ at $a_1$ would be negative since $a_2$ in expectation gives lower future reward than $a_1$. Thus both gradients are negative in this case.
> > > > > > > > > > >     * We could construct the same argument for the case where $a_2$ is better than $a_1$ and in that case both gradients would be positive.
> > > > > > > > > > >     * Thus, the 2 gradients are likely to be positively correlated in AIL.
> > > > > > > > > > >     * **If you'd like us to construct a toy environment and show this empirically, kindly let us know and we will try it out.**
> > > > > > > > > > > * **Statement needs to be reversed**: Thank you for pointing this out. We will update it.
> > > > > > > > > > > * **SNR as a proxy for gradient variance**: SNR is a measure of signal to noise ratio. Higher SNR means a cleaner signal. Lower SNR means a noisier signal. Thus intuitively, if gradient's SNR is high then the gradient is less noisy and hence is a better learning signal. SNR has also been used in the literature to analyze policy gradient algorithms. Roberts, J.W. and Tedrake, R., 2008, January. Signal-to-Noise Ratio Analysis of Policy Gradient Algorithms. In NIPS. Link: https://proceedings.neurips.cc/paper/2008/file/8df707a948fac1b4a0f97aa554886ec8-Paper.pdf
> > > > > > > > > > >     * We hope that you are convinced SNR is a good proxy. If not, kindly suggest us alternative approach(es) to perform this analysis that you'd like us to try.

---

> > > > > > > > > > > > ### Comment · Reviewer_AjRj · 2022-08-21
> > > > > > > > > > > > **Response**
> > > > > > > > > > > >
> > > > > > > > > > > > You substantiate the claim that the gradients are positively correlated on a bunch of further unsubstantiated claims, which is not useful. Most of these points miss, that the reward function of GAIL is computed based on probability densities of both, the expert and the agent. For example, if the agent follows an expert trajectory it will not in general get the highest reward, it could
> > > > > > > > > > > > get penalized for it, if the agent always use that same trajectory. More importantly, the discriminator may predict very high reward far from the expert demonstrations due to function approximation errors. For empirical evidence, you should not construct a toy problem, but rather estimate the SNR on the actual (MuJoCo) experiments. In any way, I suggest to soften the claims unless you can provide convincing evidence.
> > > > > > > > > > > >
> > > > > > > > > > > > However, I agree that signs of gradients only need to match in expectation and that the provided reference is a convincing argument for investigating the SNR (please add the reference to the paper).

---

> > > > > > > > > > > > > ### Author Response · Authors · 2022-08-25
> > > > > > > > > > > > > **Response**
> > > > > > > > > > > > >
> > > > > > > > > > > > > Thank you for the feedback.
> > > > > > > > > > > > >
> > > > > > > > > > > > > Kindly take a look at the revision: https://openreview.net/forum?id=V3Mjpi4kzdn&noteId=vx6_73pWIV (changes colored in blue)
> > > > > > > > > > > > >
> > > > > > > > > > > > > 1. **Softened claims**: In section 4.5, instead of saying "*ARC is likely to outperform AC*", we say "*we perform SNR analysis and derive the conditions under which ARC is likely to outperform*" & "*we believe that AIL is likely to present favorable conditions for ARC*".
> > > > > > > > > > > > > 1. **Added reference to SNR paper**
> > > > > > > > > > > > >
> > > > > > > > > > > > > Kindly let us know if the changes look good.

---

> > > > > > > > > > > > > > ### Author Response · Authors · 2022-08-26
> > > > > > > > > > > > > > **Request for Feedback**
> > > > > > > > > > > > > >
> > > > > > > > > > > > > > Dear Reviewer,
> > > > > > > > > > > > > >
> > > > > > > > > > > > > > As the deadline is approaching, kindly let us know if you are satisfied with the changes or if you have any additional comments.
> > > > > > > > > > > > > >
> > > > > > > > > > > > > > **Looking forward to hearing from you**.

---

> ### Author Response · Authors · 2022-08-18
> **Response to (2) Overcomplication**
>
> Thank you for pointing out this simplicity. We had added Theorem 1 for completeness. Theoretically it is the same as parameterizing Q network as r+C. However, explicitly writing the updates in terms of C makes the equations cleaner and easier to implement.
>
> Based on your recommendation, we have pushed Theorem 1 to the appendix.

---

> ### Author Response · Authors · 2022-08-18
> **Response to (3) Blurring the Contribution**
>
> We agree with you that the reward in AIL is shaped/dense as it comes from an adversary. The adversary reward is a measure of how close the immediate agent action is to the expert’s action.  Based on your recommendation, we have mentioned this in section 4.5 in blue text.

---

> ### Author Response · Authors · 2022-08-18
> **Response to "when decomposition does not help"**
>
> In Appendix A2, we have shown that ARC leads to more accurate estimates of Q gradient w.r.t. policy parameters assuming:
>
> 1. We have a correct differentiable reward function.
> 1. The error in gradient due to function approximation of Q and C networks are similar.
>
> ARC outperform AC if these assumptions are valid. ARC would under-perform AC if the error in gradient due to function approximation of $C$ network is significantly higher than that of $Q$ network. ARC might also be hurt if the reward signal is misleading, but in AIL the reward is not misleading (assuming a reasonable adversary). Based on your recommendation we have mentioned this in the paper in Section 4.5 in blue text.

---

> ### Author Response · Authors · 2022-08-18
> **Response to "actions can be difficult to obtain"**
>
> We agree that there are several scenarios where obtaining expert actions is difficult but it is still a common practice to obtain (s,a) pairs. Of all papers published in CoRL 2021, 12 papers mention “demonstration” or “imitation” in their titles. 11 out of those 12 papers use demonstration containing (s,a) pairs and only 1 paper uses (s) only. Having access to (s,a) pairs instead of (s) only adds valuable information to train a policy. We believe this is why using (s,a) pair is a popular choice, despite the difficulties associated with it.
>
> We have updated the limitations section to mention that there are several scenarios where obtaining (s,a) pairs is difficult.

---

### Official Review · Reviewer_iG9x · 2022-08-03

**Originality:** Very Good
**Technical Quality:** Good
**Clarity Of Presentation:** Very Good
**Impact:** 3

**Recommendation:**

Weak Accept: I recommend accepting the paper, but will not argue for my recommendation if the majority of other reviewers have a different opinion.

**Summary:**

The authors introduce Actor Residual Critics (ARC) as an alternative to the typical critic structure used in actor-critic RL, where the residual critic learns to estimate only the discounted future reward in exclusion of the immediate reward, which is assumed to be representable by a differentiable function. To aid in imitation learning, they further use ARC with adversarial imitation learning techniques such as those pioneered by Generative Adversarial Imitation Learning (GAIL) where immediate rewards can be estimated by a neural network-based differentiable function. The authors demonstrate that their ARC-based imitation learning methods can outperform prior work in a number of continuous control tasks in both simulation and on real robots.

**Issues:**

1. Please better address how the trained agents could perform significantly better than the experts.
2. Weaken assertion that having expert state-action pairs is something that is typically reasonable to expect.
3. Offer more clarity in the main paper on how expert data is obtained and used as well as more thorough discussion (in the appendix if space doesn't allow) on the relative performances of the baseline algorithms.
4. Strengthen discussion on related works for fully differentiable model-based imitation learning as this will provide stronger context for readers.

**Quality Of The Limitations Section:**

Additional details required

**Reviewer Expertise:**

4: The reviewer is confident but not absolutely certain that the evaluation is correct

**Robotics Focus:**

Sufficient demonstration on hardware

**Strengths And Weaknesses:**

## Strengths:
- The core premise of the work is fairly clearly laid out and the paper itself is well written and relatively easy to follow, given at least a cursory understanding of the field.
- I appreciate that the authors start by demonstrating the value of their approach on a simple toy problem in section 5.1 as it helps better establish the fundamental utility of their work.
- There is relatively strong evidence to support the efficacy of the introduced method

## Weaknesses:
- The argument that maximizing the immediate differential reward in Equation 8 can result in a sub-optimal short-sighted policy seems sound to me for the general RL case, however, for the imitation learning case - particularly with methods that attempt to maximize occupancy matching, would the immediate reward not be a measure of how closely the policy matches the expert's behavior? It is curious to me that naive-diff seems to perform so poorly in such cases and furthermore, it is surprising that ARC-based techniques so significantly outperform the expert (which was presumably used for demonstration) on the ant and cheetah tasks. How is such an improvement over the expert even possible if imitation learning was the goal? These are aspects of the method, results and discussion that I feel bear more thorough examination.
- There is a rather significant drop in GAIL's performance on the Hopper task results in Fig. 4, though, unless I missed something, this isn't discussed much. Given my own experience with the algorithm and also its performance on the other tasks, the large drop is concerning and a cause ought to be clearly identified so as to rule out a fundamental issue with the training structure, reference data and/or implementation.
- ARC-AIL being limited only to training with state-action pairs is something that the authors recognize, however, I feel that this limitation is being understated -- the assumption that one typically has access to both for practical imitation learning can be a weak one, particularly if the demonstrator is not action-compatible with the trained agents, i.e. if the action spaces for the expert are different from the agent (eg. in the case of a human demonstrating a tool path for a robot to follow without directly guiding the robot). In such cases, methods that perform occupancy matching with (state, next-state) pairs (such as Adversarial Motion Priors [Peng et al., SIGGRAPH 2021]) might be more effective.
- Training with only 5 random seeds does not seem statistically strong, particularly when the RL community is moving towards improving the statistical significance of results. Results would be stronger if more tests can be conducted, though I also understand that resource and time restrictions may limit this.
- Unless I missed it, it seems that the source(s) of the expert demonstrations is never clearly addressed in the main paper. This is potentially important information for a reader to judge the the algorithm on the merits of the reported performance relative to an expert.
- While I understand that the authors seem to want to draw a clear distinction from their work to differentiable model-based inverse reinforcement learning/imitation learning, I believe more discussion on the subject is warranted in order to give audiences a clearer picture of the methods and relative performances of that class of algorithms.


**Summary Of Recommendation:**

I believe that the paper is overall sufficiently technically sound and that results are mostly convincing. However, there are a number of issues with the presentation of the work which limit my confidence in it and leave the assertions with clear vulnerabilities that perhaps ought to be better addressed.

---

> ### Author Response · Authors · 2022-08-18
> **Response**
>
> Thank you for your valuable feedback. Kindly
>
> 1. Take a look at our response to each of your concerns in separate comments.
> 1. Refer to the **Revision 1** of the paper: https://openreview.net/forum?id=V3Mjpi4kzdn&noteId=vx6_73pWIV

---

> > ### Author Response · Authors · 2022-08-20
> > **Awaiting a response**
> >
> > Dear Reviewer,
> >
> > We are awaiting your response. Kindly let us know if we have addressed your concerns or if you have any additional comments.
> >
> > Looking forward to hearing from you.

---

> > > ### Author Response · Authors · 2022-08-25
> > > **Request for feedback**
> > >
> > > Dear Reviewer,
> > >
> > > To the best of our knowledge, we have tried to address all of your concerns. Kindly let us know if we missed something or if you have any additional comments.

---

> > > > ### Author Response · Authors · 2022-08-26
> > > > **Request for feedback**
> > > >
> > > > Dear Reviewer,
> > > >
> > > > As the deadline is approaching, kindly let us know if you are satisfied with the changes or if you have any additional comments so that we can try to accommodate them in the final submission.
> > > >
> > > > **Looking forward to hearing from you**

---

> > > > > ### Comment · Reviewer_iG9x · 2022-08-27
> > > > > **Tentatively final feedback**
> > > > >
> > > > > Based on the authors' responses and the revisions made, I do feel like the quality of the paper has been improved. I am still in favor of the paper's acceptance unless one of the other reviewers can make a strong case against it.

---

> > > > > > ### Author Response · Authors · 2022-08-27
> > > > > > **Thank you**
> > > > > >
> > > > > > Thank you so much for the positive feedback.

---

> ### Author Response · Authors · 2022-08-18
> **Response to "poor performance of naive-diff" and "outperforming expert"**
>
> **(A) Response to poor performance of naive-diff**
>
> Even in imitation learning, matching just the immediate expert action is a short-sighted/suboptimal approach. E.g. Behavior Cloning which matches the immediate expert actions, is known to lead to distribution shift and poor performance at test time ([19,10] in paper). This is the reason why most existing imitation learning methods (except Behavior Cloning), including the ones the are based on occupancy matching, optimize for the full return and not just the immediate reward. This includes GAIL, AIRL, f-Max-RKL, f-IRL, etc.
>
> **(B) Response to outperforming expert**
> Although surprising, this does happen quite often with some mujoco environments and has been reported several times in well known papers such GAIL and AIRL.
> 1. [GAIL (NeurIPS 2016)](https://papers.nips.cc/paper/2016/file/cc7e2b878868cbae992d1fb743995d8f-Paper.pdf): In Figure 1, Halfcheetah, Hopper, Walker, Humanoid
> 1. [AIRL (ICLR 2018)](https://openreview.net/pdf?id=rkHywl-A-): In Table 2, HalfCheetah
> 1. [BC Augmented GAIL (CoRL 2020)](https://corlconf.github.io/corl2020/paper_25/): In Table 1, Walker2d and HalfCheetah

---

> > ### Comment · Reviewer_iG9x · 2022-08-21
> > **Response to initial response**
> >
> > (A) It is true that directly matching the immediate behavior can be a bad approach with techniques like behavior cloning, given that it's basically supervised learning. However, with the benefit of environmental interaction plus a differentiable imitation reward, I would have expected naive diff to perform better. It's just still surprising to me just how poorly it performs.
> >
> > (B) While there have been cases of the imitation agents performing better, the expert performance was typically within the variance bounds in prior work. It stood out to me that you had cases that exceeded that. However, I take your point.

---

> ### Author Response · Authors · 2022-08-18
> **Response to drop in GAIL's performance**
>
> We agree with you that GAIL in general performs well. Our choice of hyper-parameters was inspired by [28] and we tried to use the same hyper-paramters across all the environments as much as possible. We didn’t tune extensively for individual environments (both our method and the baselines). It is possible that for the Hopper environment, further tuning would stabilize. Note that the GAIL reward finally does recover and we compare the final reward (which is high).
>
> Further details on the hyper-parameter choices are presented in Appendix E2.

---

> > ### Comment · Reviewer_iG9x · 2022-08-21
> > **Response to initial response on GAIL's performance drop**
> >
> > Using the same hyperparameters across all the environments, while it has been done before, is not, in my opinion, a good enough justification. I will not hold your work individually at fault, since there have been multiple cases of it in our field, but simply using an arbitrary set of parameters, despite the fact that we know that performance can vary dramatically as a function of said parameters, feels somewhat lazy. I recognize that there are resource, time and space limits that restrict more extensive study - I have often battled the same restrictions myself - but when anomalies are observed, as with the dramatic drop in Hopper performance in your Fig 4, I think it warrants further investigation.
> >
> > What strikes me the most is that it is not a high-variance drop, especially when looking at the performance after 800k steps. My interpretation of that data is that the performance of all the seeds tested consistently collapsed between 600-800k steps and then very consistently climbed again. That seems strange. If it is indeed a function of hyperparameters, I would be more convinced if you were to demonstrate that different parameters do not contribute to the same problem and/or otherwise show that there is not a fundamental issue with how your codes and/or environments are implemented.

---

> > > ### Author Response · Authors · 2022-08-22
> > > **Tuning GAIL for Hopper**
> > >
> > > Thank you for the insightful feedback.
> > >
> > > As per your suggestion, we tuned GAIL on the Hopper environment and got a stable training curve. We have updated Figure 4 and Table 2.
> > > Kindly take a look at the revision: https://openreview.net/forum?id=V3Mjpi4kzdn&noteId=vx6_73pWIV
> > >
> > > **What was changed**: We tried different values of SAC alpha (entropy parameter), policy learning rate and learning rate schedule. Adjusting the learning rate schedule helped stabilize the training. We have mentioned this in Appendix E2 in blue text.

---

> ### Author Response · Authors · 2022-08-18
> **Response to requirement of state-action pair**
>
> We agree that there are several scenarios where obtaining expert actions is difficult but it is still a common practice to obtain (s,a) pairs. Of all papers published in CoRL 2021, 12 papers mention “demonstration” or “imitation” in their title. 11 out of those 12 papers use demonstration containing (s,a) pairs and only 1 paper uses (s) only. Having access to (s,a) pairs instead of (s) only adds valuable information to train a policy. We believe this is why using (s,a) pair is a popular choice, despite the difficulties associated with it.
>
> We have updated the limitations section to mention that there are several scenarios where obtaining (s,a) pairs is difficult.

---

> ### Author Response · Authors · 2022-08-18
> **Response to Training with more than 5 random seeds**
>
> While many papers train with 3 seeds, we used 5 seeds to train. We appreciate you understanding the resource and time limitations. We have 4 mujoco environments with 7 algorithms and 2 fetch environments with 4 algorithms. This means we have run 36*5 seeds = 180 experiments, each of which takes several hours to run. We will try our best to obtain results with more seeds but please understand the limitations.

---

> > ### Comment · Reviewer_iG9x · 2022-08-21
> > **Response to initial response on number of seeds**
> >
> > As I said, I know that getting sufficient training data can be resource intensive and that we do not all have the same access to resources to be able to get enough data within the timeline restrictions imposed by deadlines. If you are able to add more seeds in time for the final submission of this paper, I believe it will help your data credibility - particularly in cases where variance is currently high.

---

> > > ### Author Response · Authors · 2022-08-25
> > > **Training with 10 Seeds**
> > >
> > > Thank you for the feedback. Kindly take a look at the revision: https://openreview.net/forum?id=V3Mjpi4kzdn&noteId=vx6_73pWIV.
> > > (We have updated Table 2 and Figure 4)
> > >
> > > We have run the algorithms with 10 seeds. The results are similar to that those with 5 seeds.

---

> ### Author Response · Authors · 2022-08-18
> **Response to source of expert trajectories**
>
> This is mentioned in Appendix Section E Experimental Details (E2 3rd paragraph and E3 2nd paragraph).

---

> > ### Comment · Reviewer_iG9x · 2022-08-21
> > **Response to initial response on sources of experts**
> >
> > My issue was not that you did not address these in the supplementary but rather that they were simply not explained in the main paper - instead requiring readers to look into the supplementary material (which distracts from the main flow).
> >
> > I see that you have added more direction in the revision, but I would still encourage at least a brief description within the main paper to outline how your baselines were setup - at least enough so that readers would not have to divert their attention to an entirely different part of the paper or a separate document in order to find that information. I say this because I believe knowing how an experiment is setup is important in lending credibility to the results.

---

> > > ### Author Response · Authors · 2022-08-22
> > > **Response**
> > >
> > > Thank you for the valuable feedback. We agree with your point and have made changes accordingly:
> > >
> > > * We have briefly explained the source of expert trajectories and baseline implementation in Section 5.2 and 5.3 (apart from the full details in the Appendix E)
> > >
> > > * We have also added a more thorough discussion on relative performances of the baseline algorithms in Appendix F.
> > >
> > > Kindly take a look at the revision: https://openreview.net/forum?id=V3Mjpi4kzdn&noteId=vx6_73pWIV.
> > >
> > > All of the changes are highlighted in blue text.

---

> ### Author Response · Authors · 2022-08-18
> **Response to comparison with fully differentiable model-based imitation learning**
>
> We have added this is the related work in blue text and Appendix E5. We also mention this here:
>
> If we have access to a differentiable dynamics model, we can directly obtain the gradient of the expected return (policy objective) w.r.t. the policy parameters. Since we can directly obtain the objective's gradient, we do not necessarily need to use either a critic ($Q$) as in standard Actor Critic (AC) algorithms or a residual critic ($C$) as in our proposed Actor Residual Critic (ARC) algorithms.

---

### Official Review · Reviewer_FH78 · 2022-08-03

**Originality:** Good
**Technical Quality:** Good
**Clarity Of Presentation:** Excellent
**Impact:** 2

**Recommendation:**

Strong Accept: I recommend accepting the paper and will argue for my recommendation even if other reviewers hold a different opinion.

**Summary:**

This paper proposes to decompose the Q-value as immediate rewards and residual returns (i.e., the sum of rewards from t=1 to T). This decomposition makes the actor can leverage the gradient from the immediate rewards, which is likely to improve performance. The experimental results show a slight performance gain.

**Issues:**

I don't have specific questions for this paper.

**Quality Of The Limitations Section:**

Limitations are addressed clearly

**Reviewer Expertise:**

3: The reviewer is fairly confident that the evaluation is correct

**Robotics Focus:**

Relevant but unlikely to deploy to hardware in near future

**Strengths And Weaknesses:**

## Strength
The idea proposed in this paper is sensible and novel to me.

## Weakness
How residual critic performs better than the actor-critic method is still unclear to me. The major difference between ARC and AC is that ARC leverages the gradient of the immediate rewards. The author believes that the use of a gradient of immediate rewards is the primary reason for performance gain. Yet, it is hard to believe that the gradient of one step is influential.

One suggestion may be comparing the gradients in a simpler policy class to see how the gradient of ARC and AC differ from the true gradient. That said, I don't demand the author to provide this evidence during the rebuttal. This is more or less a suggestion.

**Summary Of Recommendation:**

This paper proposes a reasonable idea and shows positive results. Thus, I recommend for "Weak Accept".

---

> ### Author Response · Authors · 2022-08-18
> **Response**
>
> Thank you for your valuable feedback. Kindly refer to the Revision 1 of the paper: https://openreview.net/forum?id=V3Mjpi4kzdn&noteId=vx6_73pWIV
>
> Here are our response to the concerns raised:
> 1. **Why ARC performs better**: Both standard Actor Critic (AC) and proposed ARC estimate the gradient of the Q function to update the policy parameters. In ARC, we leverage the exact gradient of the immediate reward which is likely to help produce a more accurate estimate of the Q gradient. This provides a better training signal. This is explained in more detail in Section 4.5, Appendix A2 and empirically verified in Appendix D2.
> 1. **Compare gradient of ARC and AC**: Kindly refer to the Appendix D2 where we use a simple 1D driving environment to illustrate that ARC produces more accurate gradients than AC using a simple 1D driving environment.
> 1. **Unlikely to deploy to hardware in near future**: Kindly note that we have shown our results on a real robot. We show a real Jaco robot arm performing 2 tasks - reaching a target goal location and pushing an object to a goal location. Kindly refer to Section 5.4, Appendix D1 and supplementary slides with embedded videos.

---

> > ### Author Response · Authors · 2022-08-20
> > **Awaiting a response**
> >
> > Dear Reviewer,
> >
> > We are awaiting your response. Kindly let us know if we have addressed your concerns or if you have any additional comments.
> >
> > Looking forward to hearing from you.

---

> > > ### Comment · Reviewer_FH78 · 2022-08-23
> > > **response**
> > >
> > > Yes, the author addressed my questions. I would raise my score to "Accept." (Unfortunately, it seems that the system does not allow me to modify the score at this moment

---

> > > > ### Author Response · Authors · 2022-08-23
> > > > **Thank you**
> > > >
> > > > Thank you so much for the positive feedback!

---

### Author Response · Authors · 2022-08-18
**Revision**

**Comment:**

Kindly take a look at the attached full paper (main body + appendix) where we incorporate the suggestions of the reviewers. Changes are shown as **blue** text.

In case any reviewer missed it, we have **REAL ROBOT VIDEOS** in the supplementary slides (file named `Jaco Push Videos.pptx`)


**Zip File:**

/attachment/687faccfcd48241bab3645ff7bae01d087a82a98.zip

---

### Meta-Review · Area_Chair_vtMb · 2022-08-15

**Recommendation:** Accept (Poster)
**Confidence:** 5

**Metareview:**

While all reviewers appreciated the idea, there are severe concerns in particular from R3 concerning the validity of the statements and the contribution:
- In many standard RL settings, the  immediate reward might be misleading. The proposed algorithm seems only to have a benefit if a properly shaped reward signal is used (which is arguably the case for many standard RL benchmarks). Yet, this is not properly discussed in the paper.
- The additional theorem for the new value iteration algorithm seems to unnecessary and an overcomplication
- The contribution is mixed between RL and IRL, where in my opinion the paper makes a stronger case for the IRL setting as here the reward is always shaped. The contribution should be more focused to avoid "washing out the message".
- More seeds are needed
- Comparisons with GAIL need to be clarified (e.g. poor performance of GAIL in some environments).

The paper is currently borderline so all these issues need to be addressed very carefully.
Rebuttal Update: The paper has been improved considerably after an intensive discussion with the reviewers in the rebuttal phase. All issues have been addressed and reviewers have now a positive opinion on the paper and think it should be published.


**Best Paper Nomination:**

No

---

> ### Author Response · Authors · 2022-08-18
> **Response**
>
> Thank you for your valuable feedback. Kindly
>
> 1. Take a look at our response to each of your concerns in separate comments.
> 1. Refer to the **Revision 1** of the paper: https://openreview.net/forum?id=V3Mjpi4kzdn&noteId=vx6_73pWIV
>
> **Kindly let us know if these changes look good and if you'd like us to make additional changes**.

---

> > ### Author Response · Authors · 2022-08-25
> > **Request for Feedback**
> >
> > Dear Meta Reviewer,
> >
> > To the best of our knowledge, we have tried to address all of your concerns. Kindly let us know if our changes look good or if you have any additional comments.

---

> > > ### Author Response · Authors · 2022-08-26
> > > **Request for feedback**
> > >
> > > Dear Meta Reviewer,
> > >
> > > As the deadline is approaching, kindly let us know if you are satisfied with the changes or if you have any additional comments so that we can try to accommodate them in the final submission.
> > >
> > > **Looking forward to hearing from you**

---

> > > > ### Author Response · Authors · 2022-08-28
> > > > **Request to mitigate technical difficulty in updating score**
> > > >
> > > > Dear Meta Reviewer,
> > > >
> > > > We would like point out that reviewer FH78 wished to increase the score but couldn't modify the score due to a technical difficulty: https://openreview.net/forum?id=V3Mjpi4kzdn&noteId=K7rnJ0Lc8vb
> > > >
> > > > We would like to request you to ensure that this technical difficulty doesn't affect the paper's score.

---

> ### Author Response · Authors · 2022-08-18
> **Response to misleading reward**
>
> There are 2 rewards:
> * environment reward (i.e. the standard MDP reward)
> * adversary reward (reward obtained from adversary in AIL)
>
> It is possible that the immediate *environment reward* is misleading. However, the *adversary reward* is a measure of closeness between agent and expert actions. It naturally is never misleading (assuming a reasonable adversary). If we have an initial bad action that the expert takes to obtain a high reward later on, then the initial bad action will have a corresponding high *adversary reward*. This is because the expert took that action and the agent would get high *adversary reward* for taking the same action.
>
> In other words, the adversary reward is dense and shaped. **Based on your suggestion, we have mentioned this is in the introduction and in section 4.5 in blue text.**

---

> ### Author Response · Authors · 2022-08-18
> **Response to additional theorem**
>
> We have moved it to the Appendix.

---

> ### Author Response · Authors · 2022-08-18
> **Response to stronger case for IRL where the reward is always shaped**
>
> We agree with you. The title of the paper was already - "ARC for Adversarial Imitation Learning" and not ARC in general. Based on your suggestion, we have emphasized this further in the paper in the introduction and in Section 4.5 (Why choose ARC over AC in AIL). The additions are written in blue text.

---

> ### Author Response · Authors · 2022-08-18
> **Response to training with more than 5 seeds**
>
> We have run the algorithms with **10 seeds**. We have updated Table 2 and Figure 4.
>
> The results are similar to that those with 5 seeds.

---

> ### Author Response · Authors · 2022-08-18
> **Response to performance drop in GAIL**
>
> **New plots**: We have tuned GAIL on the Hopper environment and got a more stable training curve. We have updated Figure 4 and Table 2.
>
> **Reason for instability earlier**: Our choice of hyper-parameters was inspired by [28] and we had tried to use the same hyper-paramters across all the environments as much as possible. We hadn’t tuned extensively for individual environments (both our method and the baselines).